# ACCELERATE QUANTIZATION AWARE TRAINING FOR DIFFUSION MODELS WITH DIFFICULTY-AWARE TIME ALLOCATION

## ABSTRACT

Diffusion models have demonstrated remarkable power in various generation tasks. Nevertheless, the large computational cost during inference is a troublesome issue for diffusion models, especially for large pretrained models such as Stable Diffusion. Quantization-aware training (QAT) is an effective method to reduce both memory and time costs for diffusion models while maintaining good performance. However, QAT methods usually suffer from the high cost of retraining the large pretrained model, which restricts the efficient deployment of diffusion models. To alleviate this problem, we propose a framework DFastQ (Diffusion Fast QAT) to accelerate the training of QAT from a difficulty-aware perspective in the timestep dimension. Specifically, we first propose to adaptively identify the difficulties of different timesteps according to the oscillation of their training loss curves. Then we propose a difficulty-aware time allocation module, which aims to dynamically allocate more training time to difficult timesteps to speed up the convergence of QAT. The key component of this is a timestep drop mechanism consisting of a drop probability predictor and a pair of adversarial losses. We conduct a series of experiments on different Stable Diffusion models, quantization settings, and sampling strategies, demonstrating that our method can effectively accelerate QAT by at least 24% while achieving comparable or even better performance.

## 1 INTRODUCTION

Diffusion models have shown great power in a variety of generative tasks (Ho et al., 2020; Song et al., 2020a; Rombach et al., 2022; Chen et al., 2023b; Ruiz et al., 2023; Song et al., 2020b), such as text-to-image generation (Rombach et al., 2022; Chen et al., 2023b; Ruiz et al., 2023; Feng et al., 2024) and text-to-video generation (Chen et al., 2023a; Wu et al., 2023; Bar-Tal et al., 2024). However, one critical limitation of diffusion models is the high computational cost, which limits the practical deployment of diffusion models. This issue becomes more serious with the increase in the model size of advanced pretrained diffusion models, *e.g.*, Stable Diffusion series (Rombach et al., 2022).

To reduce the computational costs of diffusion model inference, one line of works engaged in exploring more efficient sampling strategies to reduce sampling steps (Lu et al., 2022; Liu et al., 2021; Song et al., 2020a). Recently, from an orthogonal perspective, *i.e.*, directly compressing the network of diffusion models, quantization methods (Shang et al., 2023; Li et al., 2023; He et al., 2023b; Tang et al., 2024) have been employed.

Quantization methods can be categorized into two groups, *i.e.*, post-training quantization (PTQ) and quantization-aware training (QAT). PTQ methods (Nagel et al., 2020) do not require an expensive training or finetuning process, so they are usually resource-friendly. Nevertheless, PTQ always causes significant performance degradation on diffusion models (Shang et al., 2023; Tang et al., 2024), especially in low-bit settings. In contrast, QAT methods (Esser et al., 2019) possess a strong ability to recover performance by training/finetuning network weights and quantization parameters together, but cost a large number of computational resources, especially for large pretrained models such as Stable Diffusion. To reduce the training costs of QAT on diffusion models, previous

literature has investigated from the architecture perspective. Specifically, He et al. (2023a) propose EfficientDM that incorporates the parameter-efficient finetuning method LoRA (Hu et al., 2021) into the QAT framework. However, they allocate identical importance to each timestep with uniform sampling, ignoring difficulty discrepancy across timesteps, which we find a critical factor to diffusion optimization and convergence in this paper.

Therefore, to further reduce the time costs of QAT on diffusion models from the timestep perspective, we propose a framework named **DFastQ** (Diffusion Fast QAT). Specifically, we discover that there exists a pattern of difficulty discrepancy across different timesteps by observing task loss curves of the QAT for diffusion models as presented in Fig. 1. The oscillation level of the training loss curve is adopted to estimate the difficulty. We define *difficult timesteps* as those that are oscillatory and difficult to converge, while *easy timesteps* are smooth and easy to converge. Naturally, difficult timesteps require more training time compared to easy ones. Therefore, to obtain a better overall convergence speed, we propose to dynamically allocate more training time to difficult timesteps and less to easy ones, because it can avoid wasting training time on easy timesteps that don't need so much time to reach convergence. We adaptively identify the difficulty of each timestep during the QAT training process, and employ a timestep drop mechanism to dynamically control the time allocation. The timestep drop mechanism consists of a predictor that produces a drop probability for each timestep, and a pair of adversarial losses that adjust the probability based on difficulty level. It will make difficult timesteps obtain smaller drop probabilities and thus possess more training time, and vice versa.

To summarize, our contributions are listed as follows:

1. We introduce a framework dubbed DFastQ which is the first to accelerate QAT for diffusion models from the timestep perspective, identifying and leveraging the difficult discrepancy across timesteps.

2. We propose to adaptively identify the difficulty based on the oscillation level. We propose to allocate more training time to difficult timesteps to achieve the convergence acceleration. To control the time allocation, we propose a timestep drop mechanism consisting of a drop predictor and a pair of adversarial losses.

3. Extensive experiments on various Stable Diffusion models, quantization settings and sampling strategies demonstrate that DFastQ has a generalizable ability to accelerate QAT by at least 24% while achieving comparable or even better quality.

## 2 RELATED WORK

### 2.1 MODEL QUANTIZATION

**Definition of model quantization.** Model quantization, as a crucial technique of model compression, aims to compress deep neural networks by converting model weights and activations from 32/16 bit float-point format into lower-bit formats such as INT4/INT8. Benefiting from low-bit formats, model quantization can significantly reduce both the time and memory costs of the model. The process of model quantization can be mathematically represented as follows:

$$ w_q = \text{clip} \left( \text{round} \left( \frac{w}{s} \right) + z, q_{\min}, q_{\max} \right), $$

where $s$ is the scaling factor and $z$ is the zero-point. $w$ denotes the float-point weights/activations, while $w_q$ denotes their integer representations after quantization.

### 2.2 EXISTING QUANTIZATION METHODS FOR DIFFUSION MODELS

Recently, quantization methods have been widely adopted to compress diffusion models (Shang et al., 2023; Li et al., 2023; He et al., 2023b; Wang et al., 2023; So et al., 2023; Tang et al., 2024; He et al., 2023a). They tailor the design for the characteristics of diffusion models. For example, PTQ4DM (Shang et al., 2023) proposes to collect calibration data from a skew-normal distribution in the denoising process rather than the forward process. Q-diffusion (Li et al., 2023) proposes to

collect calibration data at certain timestep intervals and split quantization for the sensitive shortcut layers of the U-Net. Most of these methods belong to PTQ, causing obvious performance degradation on diffusion models (Li et al., 2021; Tang et al., 2024). To better maintain performance, EfficientDM (He et al., 2023a) proposes a distillation-based QAT framework for diffusion models, and also leverages LoRA (Hu et al., 2021) to mitigate the significant computational costs of fully QAT. In this work, we further reduce the time costs of QAT on diffusion models from the timestep perspective.

### 2.3 RELATION WITH LOSS WEIGHTING METHODS

Some works (Choi et al., 2022; Go et al., 2024) propose loss weighting to refine the diffusion training, *i.e.*, assigning different weights of loss to different timesteps. Go et al. (2024) use the scheme of uncertainty weighting. Choi et al. (2022) propose P2-weighting based on the signal-noise ratio. Both the loss weighting and our method prioritize a subset of timesteps. We also compare to loss weighting methods to show the superiority of our method.

## 3 METHOD

In this section, we present our method DFastQ that accelerates the convergence of QAT for diffusion models by dynamically allocating training time for different timesteps according to difficulty. Firstly, we introduce preliminaries in Sec. 3.1, regarding diffusion models and QAT for diffusion models. Then, we demonstrate our observations about difficulty discrepancy. Finally, we introduce our framework DFastQ in detail.

### 3.1 PRELIMINARIES

**Diffusion models.** Diffusion models define a forward process (a Markov chain) that gradually adds the noise to the real data $\mathbf{x}_0$:

$$q\left(\mathbf{x}_t \mid \mathbf{x}_{t-1}\right) = \mathcal{N}\left(\mathbf{x}_t; \sqrt{1 - \beta_t}\mathbf{x}_{t-1}, \beta_t\mathbf{I}\right), \ t = 1, 2, .., T, \tag{1}$$

where $\beta_t \in (0, 1)$ are a series of constants that control the variance schedule. Further, this forward process can derive the nice property that $\mathbf{x}_t = \sqrt{\bar{\alpha}_t}\mathbf{x}_0 + \sqrt{1 - \bar{\alpha}_t}\boldsymbol{\epsilon}$, where $\bar{\alpha}_t = \prod_{s=1}^{t} \alpha_s$, $\alpha_t = 1 - \beta_t$, and $\boldsymbol{\epsilon} \sim \mathcal{N}(\mathbf{0}, \mathbf{I})$. Diffusion models train a neural network $\boldsymbol{\epsilon}_\theta$ to predict the noise of the noisy variable $\mathbf{x}_t$, whose training objective is:

$$\mathbf{E}_{\mathbf{x}_0, \boldsymbol{\epsilon}, t}\left[\left\|\boldsymbol{\epsilon}_\theta\left(\sqrt{\bar{\alpha}_t}\mathbf{x}_0 + \sqrt{1 - \bar{\alpha}_t}\boldsymbol{\epsilon}, t\right) - \boldsymbol{\epsilon}\right\|_2^2\right]. \tag{2}$$

In the denoising process, diffusion models generate samples by gradually denoising from a Gaussian noise $\mathbf{x}_T \sim \mathcal{N}(\mathbf{0}, \mathbf{I})$ to $\mathbf{x}_0$, along certain trajectories $p_\theta(\mathbf{x}_{t-1}|\mathbf{x}_t)$ determined by sampling strategies (Ho et al., 2020; Liu et al., 2021; Lu et al., 2022). Different sampling strategies impact the quality and style of the generated samples.

**Quantization-aware training for diffusion models.** He et al. (2023a) propose a QAT framework for diffusion models called EfficientDM, which introduces step-wise noise distillation. Specifically, the step-wise distillation aims to minimize the MSE loss between predicted noises of the full-precision model $\boldsymbol{\epsilon}_\theta$ and the quantized model $\hat{\boldsymbol{\epsilon}}_\theta$ at each denoising timestep $t$, which can be formulated as:

$$\mathcal{L} = \mathbf{E}_{t \sim \text{Uniform}(\{1,...,T\})}\left[\left\|\boldsymbol{\epsilon}_\theta\left(\mathbf{x}_t, t\right) - \hat{\boldsymbol{\epsilon}}_\theta\left(\mathbf{x}_t, t\right)\right\|_2^2\right]. \tag{3}$$

Note that $\mathbf{x}_t$ in Eq. 3 is obtained from the denoising process $p_\theta(\mathbf{x}_{t-1}|\mathbf{x}_t)$ with the full-precision model, rather than the forward process in Eq. 1.

Eq. 3 is employed to train the network weights and quantization parameters jointly, conforming to the paradigm of LSQ (Esser et al., 2019). Moreover, they also incorporate LoRA (Hu et al., 2021) into the framework to reduce computational costs. In this paper, we follow the QAT design of EfficientDM. However, we do not adopt LoRA because we found it would bring extra time costs in the single GPU training of QAT.

### 3.2 DIFFICULTY DISCREPANCY ACROSS TIMESTEPS

Previous study (He et al., 2023a) (EfficientDM) treats each denoising step as equally important, and trains them uniformly as presented in Eq. 3. It means that in the denoising process, each $\mathbf{x}_t$ will be taken as the input for QAT training, which allocates identical importance and training time to each timestep.

However, we discover that there exists a pattern of difficulty discrepancy across different timesteps by observing task loss curves of the QAT for diffusion models, as shown in Fig. 1. We can draw the following **observations**:

i) The loss value tends to decrease with respect to timestep $t$. Timestep $t$ near $T$ (the starting point of the denoising process) tends to have a smaller loss value than the one near $0$ (the endpoint).

ii) The degree of oscillation tends to decrease with respect to timestep $t$. Timestep $t$ near $T$ tends to be more oscillatory than the one near $0$.

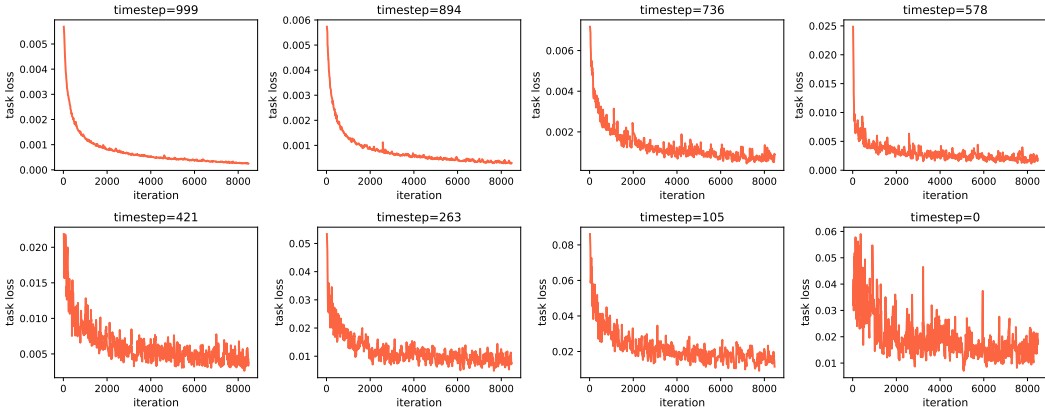

Figure 1: Task loss curves of the QAT for diffusion models, obtained with SD-2.1. $T = 1000$.

The loss value at each timestep cannot be used as an indicator to reflect the difficulty, because there is a natural difference in the order of magnitude of loss values among these timesteps. In contrast, we find that the oscillation level is a good metric to estimate the difficulty and can be easily calculated, *i.e.*, the difficulty is positively correlated with the oscillation level. We define *difficult timesteps* as those that are oscillatory, while *easy timesteps* are smooth. The principle is that oscillatory timesteps are more difficult to converge and consequently entail more training iterations than smooth timesteps.

Therefore, there is room to improve the speed of convergence by reallocating training time (training iterations) for different timesteps according to the difficulty discrepancy. Naturally, difficult timesteps require more training time compared to easy ones. To obtain a better convergence speed, we propose to allocate more training time to difficult timesteps. Note that the difficulty level is dynamic during the training process, which means we need to dynamically adjust the training time for each timestep.

### 3.3 DIFFICULTY-AWARE TIME ALLOCATION WITH TIMESTEP DROP

In this section, we elaborate on our proposed framework DFastQ, which accelerates QAT for diffusion models through dynamic training time allocation according to the difficulty discrepancy. The overall framework is presented in Fig. 2. To control the time allocation, we propose a timestep

drop mechanism whose key components are a predictor for drop probability and a pair of adversarial losses conditioned on difficulty level. The predictor outputs drop probability for each timestep, while the adversarial losses train the predictor on the basis of difficulty. We demonstrate each design in detail as follows.

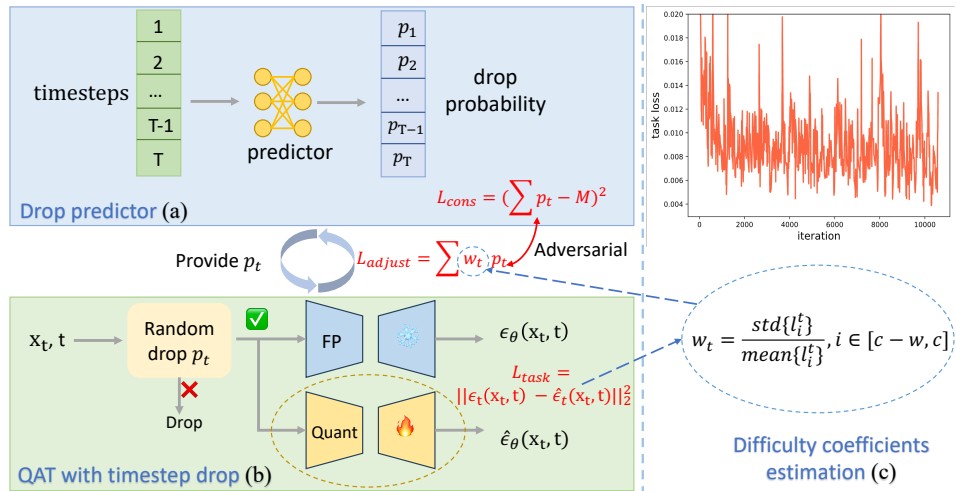

Figure 2: The overall structure of the proposed DFastQ, which consists of a predictor to output drop probability $p_t$ for each $t$, and a pair of adversarial losses $L_{cons}$ and $L_{adjust}$ to train the predictor according to the difficulty level. The QAT training progresses with the timestep drop mechanism, aiming to control the time allocation among timesteps, as more difficult timesteps tend to be assigned a lower $p_t$. We leverage the Coefficient of Variation of task loss values in the current window to estimate the difficulty level.

**Overview of the timestep drop mechanism.** The timestep drop mechanism aims to control the training time allocation for different timesteps. As stated in Sec. 3.2, ideally, the more difficult the timestep is, the more training time is allocated to it. To achieve this goal, we propose to drop timestep at random with a certain probability. Our drop mechanism ensures that difficult timesteps obtain lower probabilities than easy ones, naturally possessing more training time, and vice versa. Therefore, our method can avoid wasting training time on easy timesteps that don't need so much time to reach convergence. Consequently, the overall convergence will be faster.

**Drop predictor design.** To achieve the training time allocation for different timesteps, we need to adjust their drop probabilities dynamically. As demonstrated in recent literature (So et al., 2023; Li et al., 2023), the activation distributions between adjacent timesteps are similar. It indicates that adjacent timesteps have a similar difficulty level, which is also consistent with our observations in Sec. 3.2. Hence, inspired by So et al. (So et al., 2023), we propose to predict drop probabilities based on the timestep to better capture the relations among timesteps. Specifically, the drop predictor $\mathbf{f}_\theta$ is employed to predict the drop probability $p_t$ for each timestep $t$. It takes the embedding of $t$ as the input and outputs a scalar $p_t \in [0, 1]$, which can be easily implemented by a simple MLP network. The process can be formulated as follows:

$$p_t = \mathbf{f}_\theta(\mathbf{t_{emd}}), \quad \mathbf{t_{emd}} = \mathbf{h}_\theta(t), \tag{4}$$

where $\mathbf{h}_\theta$ is the time embedding module. Here, we reuse the frozen time embedding layers in the pretrained full-precision diffusion model as $\mathbf{h}_\theta$.

**QAT with timestep drop.** The underlying design of our DFastQ conforms to the paradigm of the well-known QAT framework LSQ (Esser et al., 2019), *i.e.*, jointly training network weights and quantization parameters. Following EfficientDM (He et al., 2023a), before each training iteration of QAT, we first collect input data $\mathbf{x}_t$ at each $t$ through the denoising process with the full-precision model $\epsilon_\theta$. Then we adopt the MSE loss as our task loss to train QAT, *i.e.*, minimizing the distance between the full-precision model $\epsilon_\theta$ and the quantized model $\hat{\epsilon}_\theta$ in terms of predicted noises at each denoising step $t$. This task loss can be written as follows:

$$\mathcal{L}_{task} = \mathbf{E}_{t \sim q(t)} \left[ \left\| \boldsymbol{\epsilon}_{\theta} \left( \mathbf{x}_t, t \right) - \hat{\boldsymbol{\epsilon}}_{\theta} \left( \mathbf{x}_t, t \right) \right\|_2^2 \right],$$

$$s.t. \ \ q(t) = \frac{1 - p_t}{\sum_{i=1}^{T}(1 - p_i)} \ \ . \tag{5}$$

Note that different from Eq. 3 of EfficientDM, $q(t)$ is no longer a uniform distribution but a distribution determined by the timestep drop mechanism. Specifically, each data point $x_t, t$ will be dropped at random with the probability $p_t$, which can be formulated as follows:

$$(x_t, t) \Rightarrow \begin{cases} train \ \hat{\boldsymbol{\epsilon}}_{\theta}, \ with \ prob. \ 1 - p_t \\ drop, \quad \quad with \ prob. \quad p_t \end{cases} . \tag{6}$$

From Eq. 6, $q(t)$ can be easily derived. Overall, timestep $t$ with a lower drop probability $p_t$ will be allocated more training time.

**Train the drop predictor based on difficulty level.** To ensure that difficult timesteps obtain lower drop probabilities than easy ones, we must train the drop predictor based on the difficulty level. As demonstrated in Sec. 3.2, the difficulty is positively correlated with the oscillation level of the task loss curve. Therefore, we use the oscillation level in the latest window to estimate the difficulty level. To quantitatively calculate the difficulty based on the oscillation, we propose a metric named *difficulty coefficient* based on *Coefficient of Variation (CV)* (Abdi, 2010), which measures the variability of the task loss value in the latest window, independently of the unit and the order of magnitude. We update the difficulty coefficient $w_t$ of timestep $t$ after every back-propagation. After $c$-th back-propagation, $w_t$ is updated as:

$$w_t = \frac{\mathrm{std}\left\{l_i^t\right\}}{\mathrm{mean}\left\{l_i^t\right\}}, \ i \in [c - w, c] \ , \tag{7}$$

where $l_i^t$ denotes the task loss at $i$-th back-propagation for denoising timestep $t$, and $w$ is the length of the latest window from which we estimate the current difficulty.

Then, we design a pair of adversarial losses to train the predictor, dynamically adjusting drop probabilities to the current difficulty coefficient. It consists of a constraint loss $\mathcal{L}_{cons}$ and an adjustment loss $\mathcal{L}_{adjust}$, which are formulated as follows:

$$\mathcal{L}_{\mathrm{cons}} = \left( \sum_{t=1}^{T} p_t - M \right)^2, \tag{8}$$

$$\mathcal{L}_{adjust} = \sum w_t p_t \ , \tag{9}$$

These two losses constitute an adversarial situation. $\mathcal{L}_{adjust}$ tries to minimize the weighted sum of $p_t$, with the difficulty coefficient $w_t$ as the weight. $\mathcal{L}_{adjust}$ tends to make every $p_t$ close to 0, but $\mathcal{L}_{cons}$ forces their sum to be a fixed hyper-parameter $M$ greater than 0. Therefore, to minimize these two losses simultaneously, $\mathcal{L}_{adjust}$ will choose to decrease the $p_t$ of the timestep $t$ that has a larger $w_t$ and increase the $p_t$ of the timestep with a smaller $w_t$. Finally, difficult timesteps (with larger $w_t$) will be assigned smaller drop probabilities, while easy timesteps (with smaller $w_t$) will be assigned higher drop probabilities. Then, the dynamic time allocation based on difficulty level through the drop mechanism is achieved. Difficult timesteps obtain smaller drop probabilities than easy ones, thus possessing more training time.

In conclusion, we alternatively train the predictor and the QAT process. The total objectives are summarized as follows:

$$\mathcal{L}_{stage1} = \min_{\mathbf{Net}_{\theta}} \left( \mathcal{L}_{task} \right), \tag{10}$$

$$\mathcal{L}_{stage2} = \min_{\mathbf{f}_{\theta}} (\mathcal{L}_{cons} + \mathcal{L}_{adjust}) \ , \tag{11}$$

where $\mathcal{L}_{stage1}$ optimizes quantization parameters and network weights of the diffusion model, i.e., $\mathbf{Net}_\theta$. $\mathcal{L}_{stage2}$ optimizes the drop predictor $\mathbf{f}_\theta$.

## 4 EXPERIMENTS

### 4.1 EXPERIMENTAL SETTINGS

**Models and datasets.** We use the Stable Diffusion v1-4 (SD-1.4) and the Stable Diffusion 2.1-base (SD-2.1) checkpoints provided by *hugging face* (hug). Following previous works (Li et al., 2023; Tang et al., 2024), we use COCO train2017 and val2017 (Lin et al., 2014) as the training and the test dataset respectively.

**Metrics.** Following Tang *et al.* (Tang et al., 2024), we adopt *FID-to-FP32* to evaluate the quality of quantized models. FID to FP32 calculates the FID distance (Heusel et al., 2017) between the generated images of full-precision models and quantized models, which can effectively measure both the image fidelity and the prompt-image matching. We do not use CLIP score (Hessel et al., 2021) because it has been shown to be poor at distinguishing the quality of quantized diffusion models (Tang et al., 2024). We generate 5,000 images using prompts from COCO val2017, and calculate the FID-to-FP32 score. We calculate the *BOPs*(Yu et al., 2020) metric to measure the theoretical computation amount of the model.

**Hyper-parameters.** In all the experiments, we maintain the same hyperparameters for our proposed modules. The learning rate of the drop predictor $\mathbf{f}_\theta$ is set to 1e-6, which is a common practice and performs well in our task. $M$ is set to $0.6T$. The window length $c$ is 5.

**The rule of determining convergence iteration.** For both baseline and our method, we test the model for every 100 iterations. We train the model until the performance deteriorates twice in a row. For example, if the FID scores of three consecutive evaluations are 10.91, 10.92 and 11.00, we will stop training, and choose the best iteration so far as the convergence iteration.

### 4.2 MAIN RESULTS

We compare our method with the QAT baseline (EfficientDM without LoRA). To validate the generalization of our method, we experiment on the settings of different model types, samplers, and quantization bit-widths. We also compare our method with state-of-the-art PTQ-based methods to demonstrate the superiority of QAT. Our metrics are all computed on the test dataset.

To begin with, we summarize the model quality at the convergence point and the total time cost by Tab. 1, where "Conv. iter." denotes the number of the convergence iterations. For baselines, an iteration means training the QAT with $S$ intermediate inputs $(x_t, t)$ in a single denoising process, where $S$ denotes the number of sampling steps. For our method, intermediate inputs in a single denoising process go through a random drop mechanism, so training with $S$ consecutive intermediate inputs is counted as an iteration. We experiment on the popular pretrained text-to-image diffusion model SD-1.4 with the default 50-step PNDM sampler (Liu et al., 2021). Besides, we also test another text-to-image diffusion model, *i.e.*, SD-2.1, with the advanced Euler (Karras et al., 2022) sampler that can generate high-quality images in 20 steps. We test three different settings of quantization bitwidth, *i.e.*, W4/A8, W5/A5, and W6/A6, where the notation "Wx/Ay" represents the bit-width of weights (W) and activations (A) respectively. The results in Tab. 1 show that our method can accelerate QAT by at least 24% while achieving comparable or even better performance. For instance, in the setting of SD-2.1 with W6/A6 bitwidth (the last row), we save the training time from 26.4h to 18.6h with the FID improvement from 15.42 to 15.08. Detailed analysis of time costs and experiments on DiT (Peebles & Xie, 2023) can be found in the Appendix.

Moreover, we plot FID-Iteration curves of the QAT training process to show the pattern of model quality change. Fig. 3 shows the results of SD-1.4, while Fig. 4 shows the results of SD-2.1. We use the stars to mark convergence points. For SD-1.4 with W4/A8 (Fig. 3a), we use the iteration 350 as the convergence point because its score (10.75) is already better than the baseline (10.91). These results show our method can not only obtain better performance at the convergence point, but also generally improve the quality during the QAT process.

Table 1: Model quality at the convergence point and the total time cost. Conv. iter. denotes the number of the convergence iteration. The training time is obtained on a single A100 GPU.

| Model type | Bits(W/A) | Method | Size (GB) | BOPs (T) | Conv. iter. | Training time | FID-to-FP32↓ |
|---|---|---|---|---|---|---|---|
| | 32/32 | FP32 | 3.44 | 693 | - | - | 0.00 |
| SD-1.4 (50-step PNDM) | 4/8 | baseline | 0.44 | 21.66 | 500 | 8.7h | 10.91 |
| | | ours | 0.44 | 21.66 | 350 | 6.5h(-25%) | **10.75** |
| | 5/5 | baseline | 0.54 | 16.92 | 1600 | 27.2h | 35.19 |
| | | ours | 0.54 | 16.92 | 1100 | 20.5h(-25%) | 36.11 |
| | 32/32 | FP32 | 3.44 | 693 | - | - | 0.00 |
| SD-2.1 (20-step Euler) | 4/8 | baseline | 0.44 | 21.66 | 2200 | 14.8h | 11.50 |
| | | ours | 0.44 | 21.36 | 1600 | 11.3h(-24%) | 11.53 |
| | 6/6 | baseline | 0.65 | 24.36 | 4000 | 26.4h | 15.42 |
| | | ours | 0.65 | 24.36 | 2600 | 18.6h(-30%) | **15.08** |

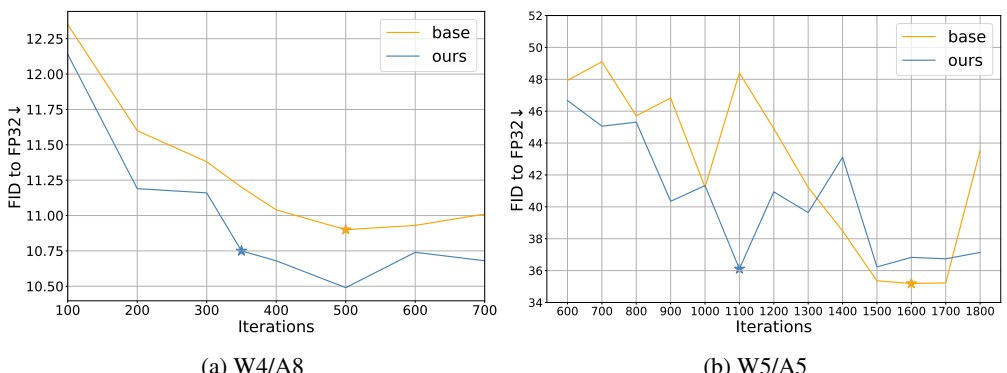

(a) W4/A8          (b) W5/A5

Figure 3: FID-Iteration curves of QAT with SD-1.4 (50-step PNDM) and various bit-width settings. A lower FID-to-FP32 score indicates better quality.

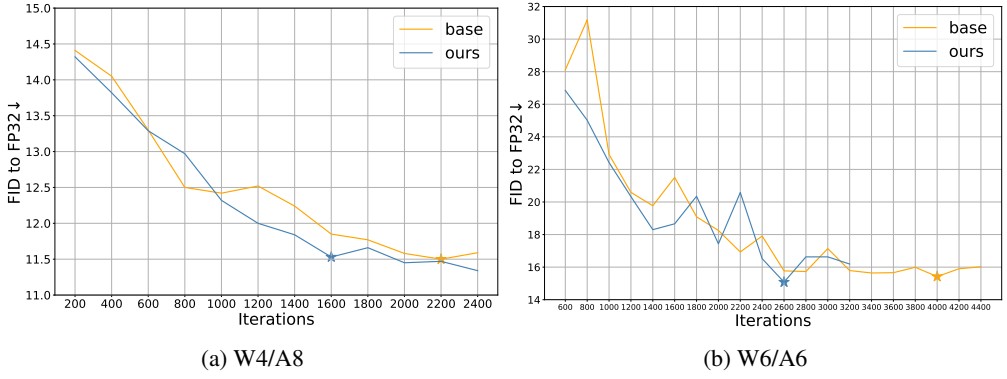

(a) W4/A8          (b) W6/A6

Figure 4: FID-Iteration curves of QAT with SD-2.1 (20-step Euler) and various bit-width settings.

Finally, Tab. 2 compares our method with state-of-the-art PTQ-based methods, regarding the performance at the convergence point and the total time cost. Our method achieves a FID-to-FP32 score of 10.75, which is a remarkable improvement compared to PTQ-based methods. Meanwhile, our method saves the training time from 8.7h to 6.5h, which is less than 7.8h of Q-diffusion and PTQ4DM, achieving a better balance between training resources and model quality.

Table 2: Comparison to PTQ-based methods. Results of SD-1.4 with 50-step PNDM sampler. The training iterations of the QAT baseline and ours are 500 and 350 respectively.

| Type | Method | Bits(W/A) | Size (GB) | BOPs (T) | Training time | FID-to-FP32↓ | CLIP score↑ |
|------|--------|-----------|-----------|----------|---------------|--------------|-------------|
| - | Pretrained FP32 | 32/32 | 3.44 | 693 | - | 0.00 | 26.46 |
| PTQ | Q-diffusion (Li et al., 2023) | 4/8 | 0.44 | 21.66 | 7.8h | 20.42 | 26.15 |
| | PTQ4DM (Shang et al., 2023) | 4/8 | 0.44 | 21.66 | 7.8h | 17.73 | 26.25 |
| | PCR (Tang et al., 2024) | 4/8 | 0.44 | 22.74 | 12.0h | 14.25 | 26.48 |
| QAT | QAT baseline | 4/8 | 0.44 | 21.66 | 8.7h | 10.91 | 26.46 |
| | DFastQ(ours) | 4/8 | 0.44 | 21.66 | 6.5h(-25%) | **10.75** | 26.46 |

## 4.3 ABLATION STUDY

**Compare to uniform drop.** We compare our method to the uniform drop where each timestep has the identical drop probability, *i.e.*, $p_t = r, \forall t$. To maintain the consistency of the sum of $p_t$, we set $r = 0.6$ as $M$ is set to $0.6T$.

**Compare to heuristic drop.** To further emphasize the significance of adaptively predicting $p_t$ based on difficulty, we compare to a stronger baseline which heuristically sets $p_t$. According to observation ii) in Sec. 3.2, the degree of difficulty (estimated by oscillation ) decreases with respect to timestep $t$. Therefore, we linearly increase $p_t$ from $0.4$ to $0.8$ with respect to $t$, which ensures the training time strictly decreases.

Fig. 5 shows the comparison of FID-iteration curves. Our method outperforms both the uniform and the heuristic methods. The curve of our method is always under others, and obtains the best FID-to-FP32 score of 10.49. The results prove the necessity of adaptively predicting drop probabilities according to difficulty level for different timesteps.

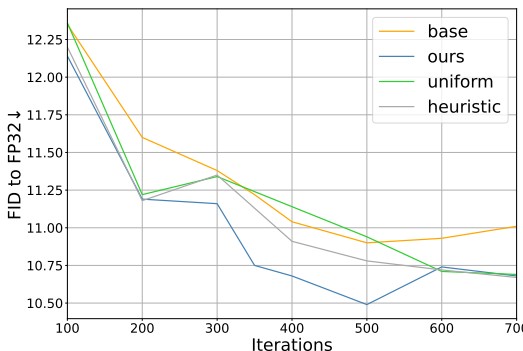

Figure 5: Ablation study. The comparison of FID-iteration curves to the uniform and heuristic methods. The experiment is conducted on SD-1.4 with 50-step PNDM sampler.

**The influence of $M$.** Tab. 3 shows the influence of the hyperparameter $M$. The results show that our method is not sensitive to M and can consistently gain faster convergence compared to the baseline.

Table 3: The ablation study of $M$ on SD 1.4 with bitwidth W4/A8

| M | 0.4T | 0.5T | 0.6T | 0.7T | 0.8T | N/A (baseline) |
|---|------|------|------|------|------|----------------|
| Conv. iter. | 350 | 300 | 350 | 400 | 300 | 500 |
| FID-to-FP32↓ | 10.90 | 10.92 | 10.75 | 10.88 | 10.83 | 10.91 |

### 4.4 Comparision to loss weighting methods

We also adapt loss weighting methods to our QAT setting, and compare our method to them, as presented in Tab. 4. Loss weight methods fail to accelerate convergence and also obtain poor quality. The reason is that they try to prioritize higher steps (steps near $T$) which are generally easy steps and only require less emphasis.

Table 4: Comparison to loss weighting methods.

| Method | Conv. Iter. | FID-to-FP32↓ |
|---|---|---|
| baseline | 500 | 10.91 |
| P2 weighting (Choi et al., 2022) | 700 | 11.50 |
| ANT-UW (Go et al., 2024) | 700 | 11.41 |
| Our | **350** | **10.75** |

### 4.5 Visual comparison

We visually compare generated images of the full-precision model, the QAT baseline, and our method. Fig. 6 shows the results of SD-1.4 and SD-2.1, which qualitatively proves that our method saves time costs without quality degradation compared to the QAT baseline.

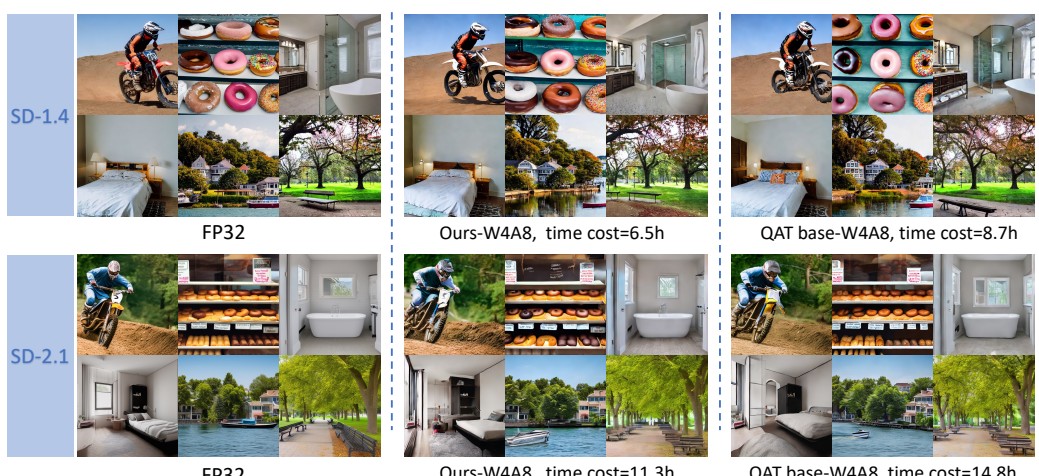

Figure 6: The visual comparison of SD-1.4 and SD-2.1.

## 5 Conclusion

In this paper, we propose a framework dubbed DFastQ to accelerate the Quantization-aware Training (QAT) for diffusion models. From the timestep perspective, we identify the difficulty discrepancy across different denoising timesteps, and propose to accelerate the convergence of QAT by allocating more training time to difficulty timesteps. To dynamically control the time allocation, we propose the timestep drop mechanism consisting of a drop probability predictor and a pair of adversarial losses conditioned on difficulty level. We conduct experiments on a variety of Stable Diffusion models, quantization settings, and sampling strategies, which proves that our method can accelerate QAT by at least 24% while maintaining the model quality.

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

# A APPENDIX

## A.1 TASK LOSS CURVES OF THE QAT.

Fig. 1 has shown the task loss curves of SD-2.1 with the 20-step Euler sampler. Here, we add the curves of SD-1.4 with the 50-step PNDM sampler, presented in Fig. A1, which also conform with the observations proposed in Sec. 3.2.

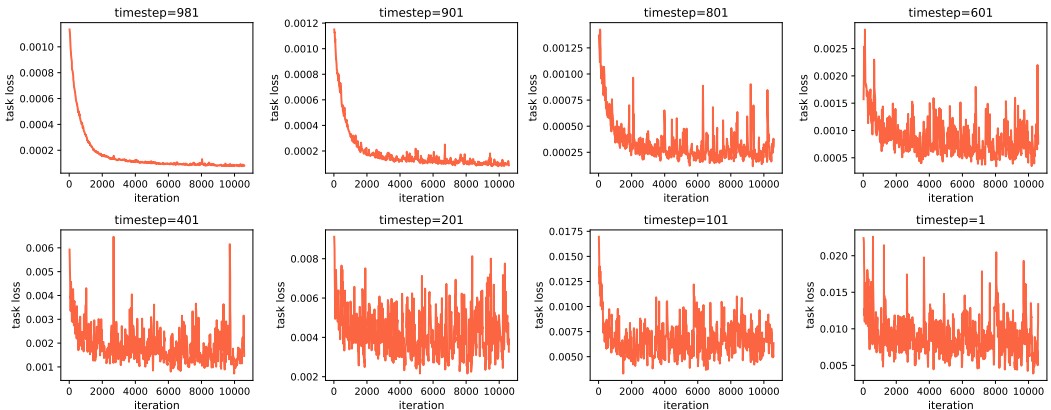

Figure A1: Task loss curves of the QAT for diffusion models, obtained with SD-1.4. $T = 1000$.

## A.2 PLOTS OF DROP PROBABILITIES AND DIFFICULTY COEFFICIENTS

Fig. A2 illustrates the curves of the drop probability $p_t$ in the training process of our framework. The results show that difficult timesteps (*e.g.*, timestep=1) are assigned lower drop probabilities, while easy timesteps (*e.g.*, timestep=981) are assigned higher ones.

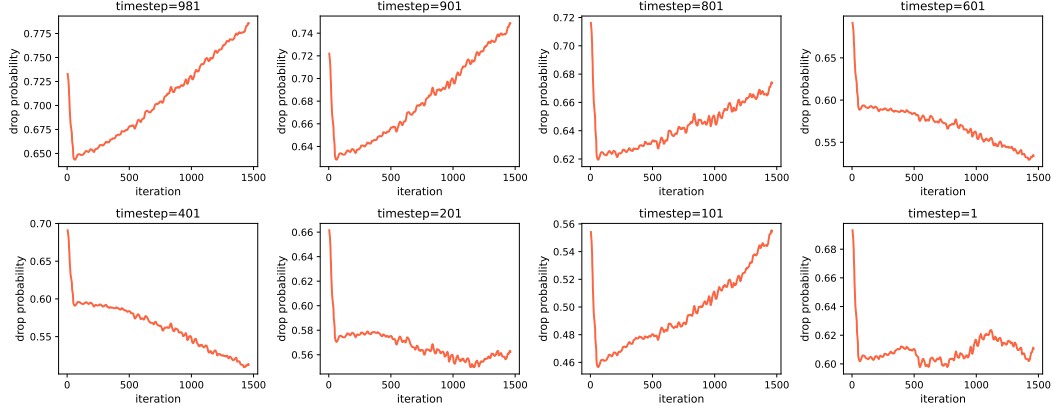

Figure A2: Drop probability curves of SD-1.4.

Fig. A3 illustrates the curves of the difficulty coefficient $w_t$. The results show that difficult timesteps (*e.g.*, timestep=1) have a larger $w_t$, while easy timesteps (*e.g.*, timestep=981) have a smaller one. It proves that the proposed difficulty coefficient is a good metric to estimate the difficulty.

## A.3 MORE IMPLEMENTATION DETAILS

We provide more implementation details of the experiments to ensure reproducibility. The learning rate of the drop predictor $\mathbf{f}_\theta$ is set to 1e-6. $M$ is set to $0.6T$. The window length $c$ is 5. Our underlying codes for QAT are built on the LSQ-Net repository. The batch size is set to 1 and AdamW optimizer

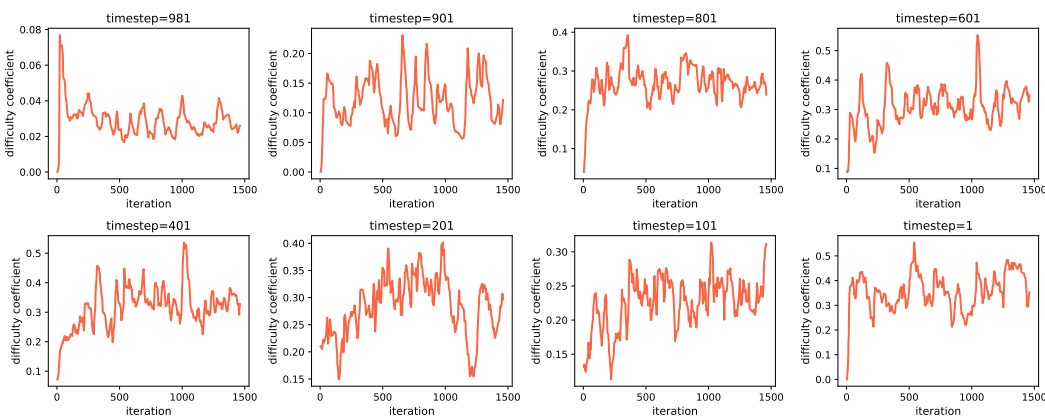

Figure A3: Difficulty coefficient curves of SD-1.4.

is employed. We use the *diffusers* package to access diffusion models and perform inference. We use the *torch-fidelity* package for fast and precise FID calculation.

## A.4 ADDITIONAL VISUAL COMPARISON

Fig. A4 provides an extra comparison to the full-precision model and the QAT baseline. The conclusion accords with Fig. 6, *i.e.*, our method can save time costs without quality degradation compared to the QAT baseline.

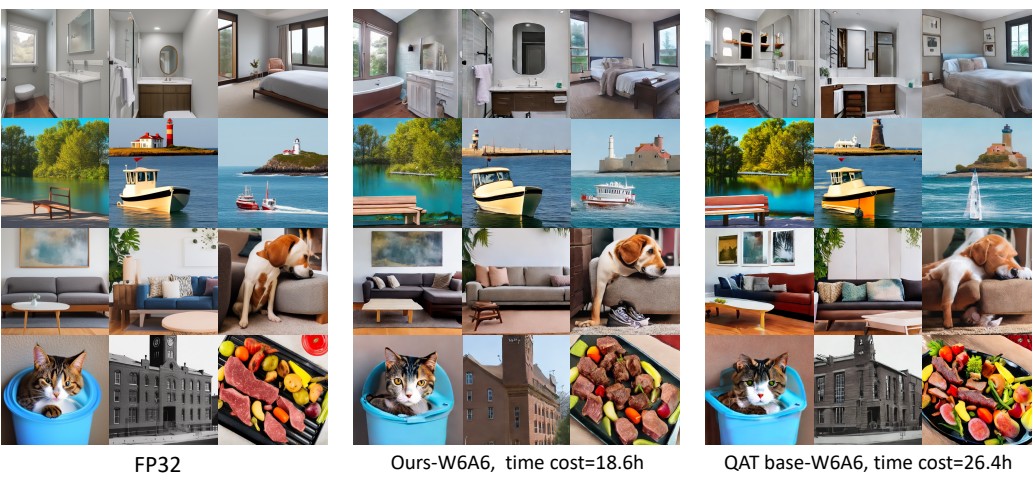

Figure A4: Additional visual comparison of SD-2.1.

## A.5 ANALYSIS OF TIME COSTS

As shown in Tab. 1, the reduction ratio of actual GPU hours is slightly less than the reduction ratio of convergence iteration, for example, 30% (from 26.4h to 18.6h) vs. 35% (from 4000 to 2600). It results from the slight extra cost of collecting more intermediate data $(x_t, t)$ through the denoising process. This extra cost is minor as the inference latency of the denoising process is much less than the QAT training time. Besides, the extra cost can be reduced by parallel inference.

## A.6 RESULTS ON DIT

The experimental results of DiT-XL/2 on ImageNet with bitwidth W4/A8 are presented in Tab. A1. We also report Inception Score following previous literature. It shows our method can save 18%

time without quality degradation. The reduction ratio is slightly lower than the SD because the DiT model is relatively smaller, which means the costs of the proposed optimization process account for more.

Table A1: The results on DiT-XL/2 with bitwidth W4/A8

| Method | Conv. Iter. | Training time | FID-to-FP32 ↓ | Inception score↑ |
|---|---|---|---|---|
| baseline | 2400 | 12.9h | 4.38 | 287.86 |
| Ours | 1800 | 10.6h (-18%) | 4.59 | 290.73 |

## A.7 VISUALIZE THE TRAINING TIME ALLOCATION

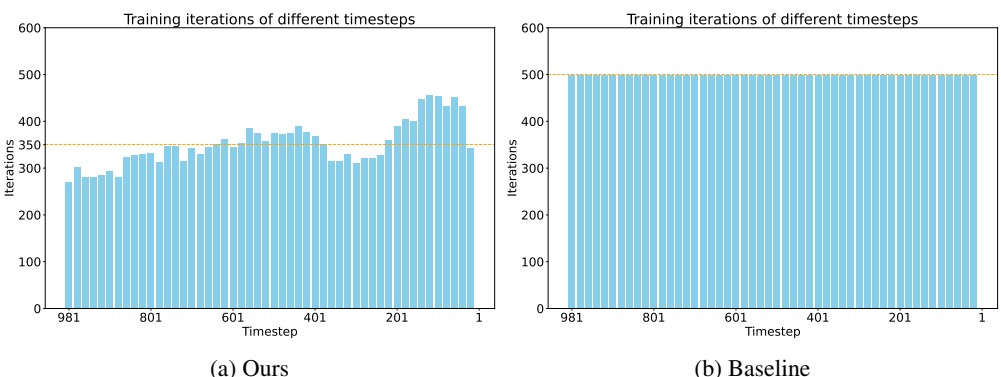

(a) Ours        (b) Baseline

Figure A5: The comparison of training time allocation between our method and the baseline. The Y-axis shows training iterations of different timesteps. The dotted line indicates the average iteration. Compared to the baseline, which allocates the same time to each step, our method adaptively allocates training based on difficulty level. The data is collected on SD-1.4 with bitwidth W4/A8.

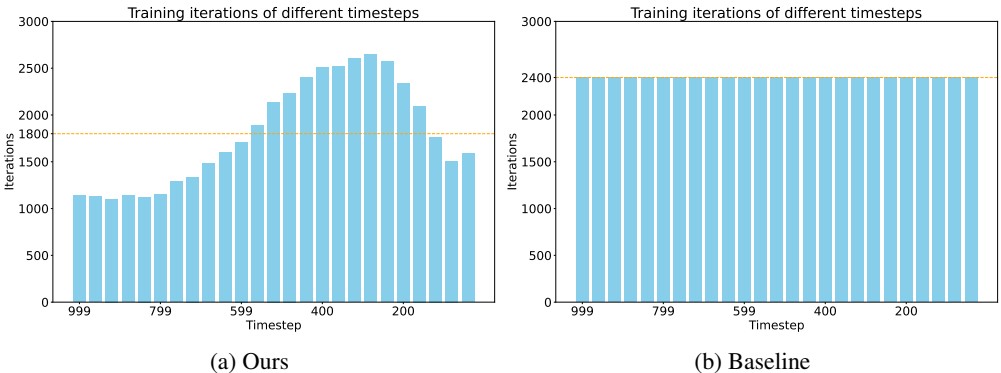

(a) Ours        (b) Baseline

Figure A6: The training time allocation of DiT-XL/2 with bitwidth W4/A8.

## A.8

## A.9 LIMITATIONS

Our experiments are on the Stable Diffusion v1-4 and v2-1 with the resolution of 512x512. It is meaningful to scale our method to larger models with larger image resolutions such as Stable Diffusion XL and Stable Diffusion 3 with the resolution of 1024x1024. QAT for these larger models naturally requires more training time, which would better emphasize the significance of our method. We consider this as our future work since our current computational resources cannot support larger experiments.

