# OpenReview forum: "Accelerate Quantization Aware Training for Diffusion Models with Difficulty-aware Time Allocation"
_ICLR.cc/2025/Conference — Submitted to ICLR 2025_

### Official Review · Reviewer_j5gh · 2024-10-29

**Soundness:** 3
**Presentation:** 3
**Contribution:** 3
**Rating:** 6
**Confidence:** 4

**Summary:**

This paper focuses on developing an efficient Quantization-Aware Training framework for the approximation and speedup of Diffusion models. The key idea behind the efficiency gains lies behind the realization that not all training timesteps are equal in terms of their impact in the final model quality, and "difficult" timesteps, where there are a lot of oscillations in the loss, are emphasized more in the proposed framework, while "easy" ones are not, thus resulting in the desired speedups.

**Strengths:**

+ The idea of selectively spending more time on difficult training timesteps cleverly and dynamically allocates the effort where needed.
+ The proposed approach is practical and appears to work well

**Weaknesses:**

- Some details are missing from the experimental evaluation (see questions below)

**Questions:**

- How well does the proposed drop predictor perform? Are there any guarantees that it will work well? What would happen to the overall performance if it does not perform well? Does it default to the uniform drop? Or can it be worse?

- In the experimental section it is stated that the same hyperparameters are maintained for all experiments. This statement is short most likely due to space limitations, but it would be important for there to be a short statement in the main body of the paper about stability with respect to hypeparameter choice.

---

> ### Author Response · Authors · 2024-11-20
> **Response to reviewer j5gh**
>
> ## Response
>
> **Thank you for the valuable comments and suggestions. We respond to your concerns point by point as follows and are glad to discuss with you.**
>
> **Q1.  The performance of the drop predictor.**
>
> Our experiments prove our proposed drop predictor performs well, which can effectively accelerate the QAT training. The principle behind this is that the component is trained under the guidance of the difficulty level, which guarantees a lower drop probability for the difficult step and a higher probability for the easy step. Our reliable designs for training objectives ensure the drop predictor performs well. Also, Figure A2 in the appendix shows the behaviours of the drop predictor in the training process, which shows it performs well.
>
> The uniform drop is an ablation study setting which doesn’t use the drop predictor but sets the drop probability to a constant. So it is not the default setting, which is just used to prove the effectiveness of our proposed components.
>
> **Q2. The hyperparameters.**
>
> The most important hyperparameter of our framework is M which controls the average drop probability. Actually, we have explored the influence of different M in Table 3. Table. 3 shows that our
> method is not sensitive to M and can consistently gain faster convergence compared to the baseline. Other hyperparameters such as learning rate is chosen just as the common practice in ML research.
>
> **Hope our response can address your concerns. Please feel free to communicate with us if you have any questions.**

---

> > ### Comment · Reviewer_j5gh · 2024-11-25
> >
> > Thank you, I acknowledge the response. It would be great if the main body of the paper was updated succinctly to reflect your responses.

---

> > > ### Author Response · Authors · 2024-11-26
> > >
> > > Thank you for your reply! According to your comment, we revise the main body of our paper to clarify these concerns. The modified parts are marked in purple (L345, L478, L313 ).

---

### Official Review · Reviewer_vQPX · 2024-11-01

**Soundness:** 1
**Presentation:** 3
**Contribution:** 2
**Rating:** 3
**Confidence:** 3

**Summary:**

This paper concerns acceleration of training or fine-tuning diffusion models (DMs), taken quantization into consideration. Such an approach can produce a good quantized model and full-precision diffusion model. The authors argue that the difficulty associated with time steps in DMs should play important role in convergence and quality of the trained DM. They propose a metric to measure step difficulty, and use this to enforce training more on hard steps. A step predictor is trained to predict difficulty level of each time step, and it can output a probability to sample a step for training. The authors further propose two losses to train the step predictor. The overall loss for training consists of two stages, one for training the quantized model and one for training the step predictor. To evaluate their method, pretrained Stable Diffusions are used to finetune on COCO dataset. EfficientDM is used as the baseline for comparison. The experimental result suggests that the proposed method seems to converge in fewer iterations with less training time and achieve comparable FID-to-FP32.

**Strengths:**

**Originality:**
- This paper propose to replace uniform step selection in traditional training methods for QAT by difficulty-based selection to train more on difficulty steps. This is reasonable. Difficulty is based on coefficient of variation which measures the occilation level of the loss.
- The authors then propose to use a neural network to predict probability to select a step for training.

**Quality:**
The results of finetuning pretrained stable diffusions to produced quantized models are encouraging. Their method can reduce much training time, compared to EfficientDM without LoRA.

**Clarity:**
The writing is quite easy to follow.

**Significance:**
The accelaration in QAT is potentially significant to practical applications.

**Weaknesses:**

The main weakness of this paper appears to be the mismatch between their model design and experimental setting.
- In order to better focus on some steps, the authors propose to predict drop probability $p_t$ for each timestep $t$. This probability belongs to [0,1] as discussed in lines 268-269. Those probabilities have a sum up to $M$, which appears in their proposed loss $L_{cons}$. However, in their experiments, $M = 0.6T$ is used and can cause $p_t >1$, since the total number of timesteps $T$ is often large (e.g. 1000). Table 3 even reports with $M=0.8T$.
- Those probabilities are used to form step selection $q(t) = (1-p_t)/\sum_j (1-p_j)$. A question arises: what it means when choosing such a large bound $M$ on the sum $\sum_j (1- p_j)$ for experiments? It seems that the authors heuristically choose $M$ in their experiments without a reasonable principle.

Beside, the authors only took EfficientDM as the main baseline, which may limit understanding about significance and applicability of their proposed QAT method.

**Questions:**

- Can the authors provide some discussion about the mismatch indicated before?
- It is unclear how to count Convergence iteration in their experiments. Can the authors explain more on it? It is very important to make a fair comparison with the baseline. One can think that the less training time may be due to early stopping, and may vanish when using the same number of iterations for those methods.

---

> ### Author Response · Authors · 2024-11-20
> **Response to reviewer vQPX**
>
> ## Response
>
> **Thank you for the constructive comments and suggestions. We respond to your concerns point by point as follows and are glad to discuss with you.**
>
> **W1 & W2. The range of the probability  $p_t$  and the clarification about the selection of M.**
>
> The reviewer might be concerned about if $p_t$ can exceed 1, and the selection about M. We think the reviewer might misunderstand some details of our method. As shown in Equation 8, we minimize  $L_\text{cons }=\left(\sum_{t=1}^Tp_t-M\right)^2$, so it means that the average drop probability of timesteps equals to $M/T$.  Therefore, M is set to the product of a constant and T, and neither large M nor large T has any effect on our framework. For example, if we set M=0.6T as in the paper, the average of $p_t$  should equal to 0.6, which is a valid probability belonging to [0,1]. Moreover, we also use a simple clip operation to ensure each probability   $p_t$  lies in a valid range, i.e., [0,1]. According to our experiments (Figure A2), these probabilities are indeed in the legal range, even without the clip operation.
>
> Besides, the ablation study in Table 3 proves that our method is not sensitive to M and can be consistently effective, and M=0.6T performs well in all our experimental settings.
>
> **W3 About the baselines.**
>
> We used EfficientDM as the main baseline because we could not find any other baselines. EfficientDM extended QAT to diffusion models for the first time, but at present no one else proposed new diffusion model QAT methods. We believe this is not because the diffusion QAT is not important, but because the technologies ( LSQ + Knowledge Distillation ) used by EfficientDM are mature and perform well. Therefore, we believe validating our method based on EfficientDM can demonstrate the significance and superiority of our method.
>
> **Q1**
>
> Please refer to W1 to W3
>
> **Q2. How to determine the convergence iteration.**
>
> Sorry for the confusion.  We clarify that our procedure to determine the convergence iteration is that: For both baseline and our method, we test the model for every 100 iterations. We train the model until the performance deteriorates twice in a row. For example, if the FID scores of three consecutive evaluations are 10.91, 10.92 and 11.00, we will stop training, and choose the best iteration so far as the convergence iteration. We make this clear in the revised paper, marked by blue (Line 347).
>
> **Hope our response can address your concerns. Please feel free to communicate with us if you have any questions.**

---

> ### Comment · Reviewer_vQPX · 2024-11-27
> **Reply to the rebuttal**
>
> Thank you for your response.
>
> I see your method use $p_t$ to select step in the training course. However, such a selection method is heuristic. The selection of one step seems to be independent with other steps, which is weird.
>
> I also read comments from other reviewers and understand that this paper seems to focus on speeding up training. In this perspective, the authors need to train a model from scratch to provide a rigorous evaluation. In the current version, the authors only finetuned a pretrained stable diffusion. Such an evaluation only provides some behaviors of their method.
>
> Since their method is mostly heuristic and the current paper is unclear about the main focus (due to lack of comprehensive evaluation), I maintain my current score.

---

> ### Author Response · Authors · 2024-11-27
>
> Thank you for your reply! We think there may be three points that need clarification.
>
> 1. Our task cannot train from scratch. The reviewer recommends us to train the model from scratch, however, our task (Quantization-aware Training, QAT) must train the quantization parameters and model weights from a pretrained checkpoint[1,2].  We aim to speed up QAT rather than the usual diffusion training.
> 2. We use  $p_t$ to select steps, where   $p_t$ is the output of the predictor taking t as the input.  [3] have demonstrated this kind of predictor can capture temporal information well. The selection of one step is not independent with other steps, because the predictor the trained with various t and can capture temporal relations between steps.
> 3. Our method is not heuristic. As presented in the paper,  $p_t$ is dynamically optimized according to the difficulty coefficient $w_t$, where $w_t$  is calculated based on the task loss values in the current sliding window (Eq.7). The overall framework is adaptive and dynamic, rather than simply heuristic.
>
> ### References
>
> [1] Learned step size quantization. ICLR’19
>
> [2] EfficientDm: Efficient quantization-aware fine-tuning of low-bit diffusion models. ICLR’23
>
> [3] Temporal dynamic quantization for diffusion models. NIPS’23

---

### Official Review · Reviewer_z9WH · 2024-11-02

**Soundness:** 3
**Presentation:** 2
**Contribution:** 2
**Rating:** 3
**Confidence:** 4

**Summary:**

This paper aims to speed up the training process of diffusion models. Quantization-aware training (QAT) is introduced to identify the difficulties of different timesteps. Then the training time is dynamically allocated more training time to difficult timesteps. Experiments on various Diffusion models are conducted to demonstrate the ability to accelerate the training process. Overall, the method is intuitive. However, the novelty is limited, and the experiment quality is lower than the bar.

**Strengths:**

1. Speeding up the training process of the diffusion model is an important research problem.
2. Quantization-aware training is introduced to identify the difficulties of different timesteps in training. Training time is dynamically allocated in different stages.

**Weaknesses:**

1. The contribution is marigal. Basically, there is only one contribution: identify the difficulties of training with QAT. Identifying different stages [1,2,3] of training is not novel.

2. According to the experimental results, there is only a 25% reduction in training time. Is it worth it to introduce such a heavy framework? A simple sampling strategy [4,6, 7] can significantly reduce the training time. The author did not compare a vast majority of works [4,5,6,7] in speeding up the training process of DM.

3. The rationale behind allocating more training time is not clear. While the stages are identified, why not just increase the learning rate?

4. The experiment quality is low. In fact, there is only one model introduced in the experiment: SD-x.x. It is not conclusive at all. More different types of DM are supposed to be experimented with.

5. Paper presentation can be improved, e.g., Figure 3, 4, and and 5 are unprofessional.

 [1] Towards Faster Training of Diffusion Models: An Inspiration of A Consistency Phenomenon

[2] Hongkai Zheng, Weili Nie, Arash Vahdat, Kamyar Azizzadenesheli, Anima Anandkumar: Fast Sampling of Diffusion Models via Operator Learning. ICML 2023: 42390-42402

[3] A Closer Look at Time Steps is Worthy of Triple Speed-Up for Diffusion Model Training

[4] Shivam Gupta, Ajil Jalal, Aditya Parulekar, Eric Price, Zhiyang Xun: Diffusion Posterior Sampling is Computationally Intractable.

 [5] Tae Hong Moon, Moonseok Choi, EungGu Yun, Jongmin Yoon, Gayoung Lee, Jaewoong Cho, Juho Lee: A Simple Early Exiting Framework for Accelerated Sampling in Diffusion Models.

[6] Zhiwei Tang, Jiasheng Tang, Hao Luo, Fan Wang, Tsung-Hui Chang: Accelerating Parallel Sampling of Diffusion Models.

 [7] Andy Shih, Suneel Belkhale, Stefano Ermon, Dorsa Sadigh, Nima Anari: Parallel Sampling of Diffusion Models. NeurIPS 2023

**Questions:**

N/A

---

> ### Author Response · Authors · 2024-11-20
> **Response to reviewer z9WH**
>
> ## Response
>
> **Thank you for the constructive comments and suggestions. We respond to your concerns point by point as follows  and are glad to discuss with you.**
>
> **W1. Clarification about our contributions.**
>
> We’d like to clarify that our contributions are two-fold:(i) We propose to automatically and dynamically identify the difficulties of training with QAT. (2) We propose a drop mechanism to achieve fast QAT training based on the dynamic difficulty. Although previous works[1,3] (the reviewer mentioned) and [8,9] (which we have compared with in the section 4.4) have explored accelerating diffusion training based on difficulty, they are either static or heuristic, while our method can **dynamically and automatically** measure the difficulty of different stages in the training process. Dynamically measuring the difficulty is important, because the difficulty varies with the current training state. Figure A2 can prove this, where the drop probability of timestep=201 decreases at first and then increases. Moreover, actually we have done experiments that compare our method to previous works (i.e., P2-weighting [8] and ANT-UW [9]) that identify the difficulty,  as shown in Table 4.  We also add a comparison with the suppression sampling in [3] (implement it in the QAT framework). We summarize the comparison to these difficulty-aware methods [3,8,9], emphasizing the superiority and contribution of our method.
>
> | Method | Convergence Iter. | FID-to-FP32 ↓ |
> | --- | --- | --- |
> | baseline | 500 | 10.91 |
> | Asymmetric sampling [3] | 700 | 11.46 |
> | ANT-UW [9] | 700 | 11.41 |
> | P2 weighting [8] | 700 | 11.50 |
> | Ours | **350** | **10.75** |
>
> **W2. Comparison to fast sampling methods[4,5,6,7].**
>
> The reviewer hopes us compare our method to these fast sampling methods[4,5,6,7], however, we are very confused about why it is required. Our task is to accelerate the training of QAR, but [4,5,6,7] aim to accelerate the sampling process and don’t involve training. We guess the reviewer may hope us to compare to fast training methods, which we present in W1. Besides, our method is easy to implement and not heavy, which only requires less than 40 lines of code.
>
> **W3. The rationale behind allocating more training time.**
>
> - After identifying the difficulty, just increasing the learning rate will result in unfairness, so it is not reasonable. Specifically, in deep learning, we usually don't know the best learning rate unless we try it one by one. We can’t ensure assigning a higher learning rate to some timesteps can make them better. So we set the learning rate fixed, which is also used by previous works that the reviewer mentioned [1,3].
> - **The rationale behind allocating more training time.** In contrast, we choose to allocate more training time to difficult timesteps. Naturally, difficult timesteps require more training time compared to easy ones. To speed up the overall convergence, our method avoids wasting training time on easy timesteps that don’t need so much time to reach convergence. For example, assuming that there are 3 different difficulty levels which require training time of 1, 2, 4  hours to achieve convergence. The baseline needs 4x3=12 hours totally (allocating the same training time for each) , while ideally our method only needs 1+2+4=7 hours by adaptive time allocation.  To visualize this, Figure A5 in the revised paper compares the training time allocation between our method and the baseline.
>
> **W4. Experiments with more models.**
>
> In the appendix (Section A.6), we have reported the results with the model DiT-XL/2. In the rebuttal, we add the experiments for EDM[10]. We validate our method on FFHQ dataset using the EDM  model from its official repository. The results with bitwidth W4/A8 are presented in the following table.
>
> | Method | Convergence Iter. | Training time | FID ↓ |
> | --- | --- | --- | --- |
> | baseline | 1300 | 7.6h | 4.61 |
> | Ours | 1000 | 6.4h (-16%) | 4.63 |
>
> The reduction ratio of EDM is lower than SD because the EDM model is relatively smaller, which means the costs of drop predictor are relatively larger. But we believe we can improve it by using a smaller predictor.
>
> **Hope our response can address your concerns. Please feel free to communicate with us if you have any questions.**

---

> > ### Author Response · Authors · 2024-11-20
> > **References**
> >
> > ## References
> >
> > [1] Towards Faster Training of Diffusion Models: An Inspiration of A Consistency Phenomenon
> >
> > [2] Hongkai Zheng, Weili Nie, Arash Vahdat, Kamyar Azizzadenesheli, Anima Anandkumar: Fast Sampling of Diffusion Models via
> > Operator Learning. ICML 2023: 42390-42402
> >
> > [3] A Closer Look at Time Steps is Worthy of Triple Speed-Up for Diffusion Model Training
> >
> > [4] Shivam Gupta, Ajil Jalal, Aditya Parulekar, Eric Price, Zhiyang Xun: Diffusion Posterior Sampling is Computationally Intractable.
> >
> > [5] Tae Hong Moon, Moonseok Choi, EungGu Yun, Jongmin Yoon, Gayoung Lee, Jaewoong Cho, Juho Lee: A Simple Early Exiting Framework for Accelerated Sampling in Diffusion Models.
> >
> > [6] Zhiwei Tang, Jiasheng Tang, Hao Luo, Fan Wang, Tsung-Hui Chang: Accelerating Parallel Sampling of Diffusion Models.
> >
> > [7] Andy Shih, Suneel Belkhale, Stefano Ermon, Dorsa Sadigh, Nima Anari: Parallel Sampling of Diffusion Models. NeurIPS 2023
> >
> > [8] Perception Priortized Training of Diffusion Models, CVPR 2022
> >
> > [9] Addressing Negative Transfer in Diffusion Models, Neurips 2023
> >
> > [10] Elucidating the Design Space of Diffusion-Based Generative Models, Neurips 2022

---

> > ### Comment · Reviewer_z9WH · 2024-11-21
> > **Response to rebuttal**
> >
> > Thanks for the detailed rebuttal. I still have concerns about the weakness.
> > W1: Regarding the novelty, my concern is that Quantization Aware is not novel. Identifying different stages [1,2,3] has already been proposed in related works. The two-fold contribution is still one contribution to me: Quantization awareness for the diffusion model
> > W2: In the abstract, the paper states that "Quantization-aware training (QAT) is an effective method to reduce both memory and time costs for diffusion model". If the final target is to reduce the cost of the diffusion model, a comparison with fast training methods [1,2,3] is necessary, because these works can reduce the training time significantly. In the response to W4, the training time only dropped 16%. It is really a marginal improvement. A simple tuning strategy of the learning rate can achieve the improvement.
> > W3: The author should use case studies to illustrate the rationale.
> > W4: The rebuttal further confirms my concern. If you look at [1], they achieve 3× acceleration of diffusion model. An improvement of -16% in running time is marginal.
> >
> > I understand the author wants to focus on QAT. But from the perspective of speeding up the diffusion model, there is still room to improve the quality and contribution of this paper.

---

> > > ### Author Response · Authors · 2024-11-22
> > > **Response to "Response to rebuttal"**
> > >
> > > **Thank you for your reply!**
> > >
> > > **W1.** We understand your concern that identifying different stages is not novel, which has been explored in [1,2,3,8,9]. But we want to emphasize that identifying difficulty in these works is not automatic and dynamic, and fails to work well in the QAT scenario, as shown in the following table.  We believe our dynamic framework has its own contribution by providing a more adaptive framework tailored to QAT needs. Comparisons with [8,9] are included in the original paper, while [3] is an extra experiment based on the reviewer’s comment.
> > >
> > > | Method | Convergence Iter. | FID-to-FP32 ↓ |
> > > | --- | --- | --- |
> > > | baseline | 500 | 10.91 |
> > > | Asymmetric sampling [3] | 700 | 11.46 |
> > > | ANT-UW [9] | 700 | 11.41 |
> > > | P2 weighting [8] | 700 | 11.50 |
> > > | Ours | **350** | **10.75** |
> > >
> > > **W2.** Thanks for your explanation. Actually, we have compared to fast training methods[3,8,9] in the above Table. These works mentioned in W1 speed up training by identifying difficulty. W1 and W2 seem to be the same problem.
> > >
> > > **W3.** Thanks for your suggestion. We’d like to use Figure A5 and A6 in the revised paper as studies. Here we take Figure A5 for example. As defined in the paper, difficult steps are those which require more training iterations to converge, and vice versa. For example, as shown in Figure A1, timestep=981 is an easy step that converges soon, while timestep=1 converges slowly. Our rationale is that: our method avoids wasting training time on easy timesteps that don’t need so much time to reach convergence.  Figure A5 visualizes the comparison of training time allocation between our method and the baseline. The baseline trains each timestep equally for 500 iterations, while our method emphasizes difficult steps (near t=0).
> > >
> > > **W4.** Although previous fast training methods[1,2,3,8,9] can achieve 3x speedup, our experiments in W1 show they are not effective in our QAT scenario. Besides, although tuning the learning rate may accelerate training, the trial-and-error process of testing different learning rates is itself extremely costly. Therefore, although the acceleration ratio obtained by our method may not be very large, it is currently the only effective method in the field of QAT.

---

> > > > ### Comment · Reviewer_z9WH · 2024-12-02
> > > > **Response**
> > > >
> > > > Thanks for the detailed rebuttal. Considering the quality of this paper, I will keep the score unchanged.

---

### Official Review · Reviewer_ZPir · 2024-11-05

**Soundness:** 3
**Presentation:** 3
**Contribution:** 3
**Rating:** 6
**Confidence:** 3

**Summary:**

This work presents a new method called DFastQ that makes training diffusion models, like Stable Diffusion, faster and more efficient. This method focuses on how hard different training steps are and adjusts the training time accordingly, giving more time to the harder steps. A key part of this approach is a mechanism that predicts which steps need more attention and uses special losses to help with this training.

**Strengths:**

The paper introduces the DFastQ framework, which innovatively applies a difficulty-aware perspective to quantization-aware training (QAT). This approach identifies the varying difficulties of different timesteps based on the oscillation of their training loss curves. By dynamically allocating more training time to difficult timesteps and less to easier ones, the method optimizes training efficiency.

The paper introduces a clever timestep drop mechanism that leverages adversarial loss functions to dynamically adjust how training time is allocated based on the difficulty of each timestep. This innovative approach not only improves the training process but also provides a sophisticated way to fine-tune model performance.

The authors provide experimental validation across multiple models and quantization settings, demonstrating the effectiveness of their proposed method. The results show that DFastQ can accelerate QAT by at least 24% while achieving comparable or improved performance metrics, such as FID scores.

**Weaknesses:**

An in-depth explanation of how the Coefficient of Variation (CV) is computed in practice—specifying the window size and how the loss values are selected—would allow for better reproducibility and understanding of how difficulty is quantified.

It would be valuable to analyze the impact of the difficulty-aware time allocation versus a uniform allocation of training time. Similarly, evaluating the effectiveness of the timestep drop mechanism independently would clarify what aspects of the proposed approach are most beneficial.

Expand the description of how the difficulty of timesteps is assessed, particularly the criteria used to define "difficult" versus "easy" timesteps.

Additionally, provide a more thorough analysis of the computational efficiency gains achieved through the proposed method.

**Questions:**

How to ensure that the difficulty assessment of timesteps remains accurate throughout the training process? Could you provide more details on the frequency and method of updating the difficulty metrics during training?

Can you elaborate on how the timestep drop mechanism affects the overall convergence speed? Specifically, how do you measure the effectiveness of this mechanism compared to traditional QAT approaches?

In Table 1, while you report improved performance and reduced training time, could you provide insights on any potential trade-offs in model quality, especially with different quantization settings?

You tested various quantization bit-widths. Could you provide more detailed findings on how the choice of bit-width impacts both the training time and the final model performance?

---

> ### Author Response · Authors · 2024-11-20
> **Response to reviewer ZPir**
>
> ## Response
>
> **Thank you for the constructive comments and suggestions. We respond to your concerns point by point as follows  and are glad to discuss with you.**
>
> **W1. Details about the Coefficient of Variation.**
>
> The Coefficient of Variation for timestep $t$ is calculated based on the latest sliding window of task losses at timestep t, as shown in Eq.(7). The task loss is defined in Eq.(5), which is the MSE loss between the original noise prediction and the quantized noise prediction.
>
> In the experiments, we set the window size to 5, as wrote in the Line 346. Experiments show it works well for various models, sampling strategies, and quantization settings.
>
> **W2. Analysis versus uniform allocation, and evaluating the effectiveness of the timestep drop mechanism independently.**
>
> - **Analysis versus uniform allocation.** Firstly, we clarify the difference between our difficulty-aware time allocation and the uniform allocation. Specifically, difficult timesteps require more training time compared to easy ones. To speed up the overall convergence, our method avoids wasting  training time on easy timesteps that don’t need so much time to reach convergence . For example, assuming that there are 3 different difficulty levels which require training time of 1, 2, 4  hours to achieve convergence. The uniform allocation  needs 4x3=12 hours totally (allocating the same training time for each) , while ideally our method only needs 1+2+4=7 hours by adaptive time allocation.  Moreover, to visualize this, we use Figure A5 in the revised paper to compare the training time allocation between our method and the uniform allocation, which proves the superiority of our method.
> - **Evaluating the effectiveness of the timestep drop mechanism independently.** In the ablation study (Section 4.3), our adaptive drop mechanism outperforms the uniform drop and heuristic drop, which we believe can prove the effectiveness and rationality of our proposed drop mechanism.
>
> **W3. Expand the description of difficulty.**
>
> In the original paper, we define difficult timesteps as those that are oscillatory and difficult to converge, while easy timesteps are smooth and easy to converge. To quantitatively measure the difficulty level, we propose a difficulty coefficient  $w_t$ , as defined in Eq.(7). Therefore, timesteps with higher $w_t$  are difficult, and those with lower $w_t$  are easy.
>
> **W4 & Q2. The analysis of the computational efficiency gains achieved through the proposed method. How the timestep drop mechanism affects the overall convergence speed.**
>
> We think this question can be answered with W2's response. Our method avoids wasting training time on easy timesteps that don’t need so much time to reach convergence, so it can gain efficiency.  Moreover, we recommend the reviewer to refer to Figure A5, which visualizes how our method saves training time by time allocation versus the baseline.
>
> **Q1. Details about the frequency and method of updating the difficulty metrics during training.**
>
> To ensure that the difficulty assessment of timesteps remains accurate throughout the training process, our method adopts a dynamic updating mechanism as presented in the paper. Specifically, “dynamic” means that after every back-propagation, we add the task loss gained from this to the sliding window, and remove the oldest loss value. Then the  $w_t$ is calculated using the losses in the updated sliding windows. We simply use a FIFO queue to implement the process. We clarify this in the revised paper, marked by orange (Line 291).
>
> **Q3. Insights on potential trade-offs in model quality.**
>
> We may not understand what the reviewer means by “tradeoff”.  We’d appreciate it if you could make it clear. By far, We don't seem to observe an obvious trade-off in our framework.
>
> **Q4.  Findings on how the choice of bit-width impacts both the training time and the final model performance.**
>
> Frankly, according to our experiments, we find no obvious correlation between the bit width and the final model performance. But bit-width seems to affect the overall training time. Low bit-width makes the training more difficult and therefore tends to lengthen the training time.
>
> **Hope our response can address your concerns. Please be free to communicate with us if have any questions.**

---

### Meta-Review · Area_Chair_u86K · 2024-12-16

**Metareview:**

There are four review comments, two reviewers gave 6 (marginally above the acceptance threshold), and two gave 3 (reject). The concerns of the reviewers include the marginal contribution, the mismatch between their model design and experimental setting, and the lack of depth analysis on the model and the experiment result. After discussion, all the reviewers decide to keep their initial score unchanged.

**Additional Comments On Reviewer Discussion:**

After discussion, the concerns of two reviewers with negative scores remain, and they keep their scores unchanged.

---

### Decision · Program_Chairs · 2025-01-22

Reject